# Gaps to Best Practices for Teleconsultations Performed by General Practitioners: A Descriptive Cross-Sectional Study

**DOI:** 10.3390/ijerph19106220

**Published:** 2022-05-20

**Authors:** Alexandre Carrier, Karyne Fernez, Jan Chrusciel, David Laplanche, Clément Cormi, Stéphane Sanchez

**Affiliations:** 1Pôle Territorial Santé Publique et Performance des Hôpitaux Champagne Sud, Centre Hospitalier de Troyes, 10000 Troyes, France; alexandre.carrier@hotmail.fr (A.C.); jan.chrusciel@hcs-sante.fr (J.C.); david.laplanche@hcs-sante.fr (D.L.); stephane.sanchez@hcs-sante.fr (S.S.); 2General Practice Department, University of Reims Champagne-Ardenne, CEDEX, 51095 Reims, France; karyne.fernez@gmail.com; 3LIST3N/Tech-CICO, Troyes University of Technology, CEDEX, 10300 Troyes, France; 4University Committee of Resources for Research in Health (CURRS), University of Reims Champagne-Ardenne, CEDEX, 51095 Reims, France

**Keywords:** telemedicine, remote consultation, guideline adherence, rural health, urban health, health services

## Abstract

The COVID-19 pandemic led to large increases in telemedicine activity worldwide. This rapid growth, however, may have impacted the quality of care where compliance with guidelines and best practices are concerned. The aim of this study was to describe the recent practices of a telemedicine activity (teleconsultations) and the breaches of best practice guidelines committed by general practitioners (GPs) in the Greater Eastern Region of France. A cross-sectional study was conducted using a 33-item questionnaire and was provided to the Regional Association of Healthcare Professionals, Union Régionale des Professionnels de Santé (URPS) to be shared amongst the GPs. Between April and June 2021, a total of 233 responses were received, showing that (i) by practicing telemedicine in an urban area, (ii) performing a teleconsultation at the patient’s initiative, and (iii) carrying out more than five teleconsultations per week were factors associated with a significantly higher level of best practices in telemedicine. All in all, roughly a quarter of GPs (25.3%, *n* = 59) had a self-declared good telemedicine practice, and the rules of good practice are of heterogeneous application. Despite the benefits of learning on the job for teleconsultation implementation during the COVID-19 lockdowns, there may be a clear need to develop structured and adapted telemedicine training programs for private practice GPs.

## 1. Introduction

Quality of care is defined by the World Health Organization (WHO) as the “degree to which healthcare services for individuals and populations increase the likelihood of desired health outcomes” [1]. Its distinction involves effectiveness, safety, and engagement in people-centered services [2]. Improving quality of care at a national level implies the continuous quality monitoring of services in order to define health policies that are adapted to and essential for reaching universal healthcare through high quality primary care [3]. The rapid introduction of innovative healthcare practices in primary care situations risks reducing quality of care when a healthcare professional (HCP) has not been properly educated or trained for such innovations [4,5]. This has been particularly relevant in the field of digital health and particularly in telemedicine for which several guidelines have been previously published [6].

The population of practicing physicians, particularly general practitioners (GPs) were mostly untrained in implementing telemedicine activities when the initial COVID-19 pandemic lockdowns started and forced them to initiate video teleconsultations with their patients [7,8]. In France, the number of teleconsultations increased from fewer than 10,000 per week in early March 2020 to 80,000 in the week of 16 March 2020 (the first week of the first national lockdown) [9]. For the year 2020, a total of 19 million teleconsultations were performed nationally, which accounted for one-quarter of all consultations that year, of which 80% of them were carried out by GPs [10]. In this context, we sought to investigate whether GPs in France were adequately educated and aware about the changes that were made in their routine healthcare delivery. The objective of this study was to describe the practices of teleconsultations and the breaches of best practice guidelines committed by GPs in the Greater Eastern Region of France.

## 2. Materials and Methods

### 2.1. Design and Participants

An online, voluntary, anonymous, and cross-sectional study was conducted among GPs working in the Greater Eastern Region of France (Région Grand-Est) from April to June 2021. No other inclusion criteria were applied. The only exclusion criterion was for GPs that had not practiced active primary care since the beginning of the COVID-19 pandemic in 2020.

The questionnaire was communicated in three ways. First, it was communicated by the Regional Association of Healthcare Professionals, Union Régionale des Professionnels de Santé (URPS). Second, the municipal and regional medical boards were asked to share the questionnaire to all registered GPs and third, it was made available on the Facebook group of a GP organization called Union Française pour une Médecine Libre (UFML). The survey was shared online using GoogleForms^®^. A reminder to complete the questionnaire was sent to the participants at 2 weeks and once again at 3 weeks after the original posting.

We used a 33-item questionnaire to describe the knowledge and quality control practices of telemedicine use by GPs. Based on a literature review of best practices for teleconsultation and recommendations of the French National Authority for Health, Haute Autorité de Santé, (HAS) [11], the questionnaire was developed by a panel of three experts including two private practice GPs (one of whom had an additional university degree in telemedicine and eHealth) and a public health physician. All were chosen for their expertise in general practice, telemedicine, methodology, and/or statistics. The questionnaire asked about respondents’ characteristics (10 items), their practices regarding teleconsultation (15 items), and about their general knowledge about teleconsultations (8 items). The questionnaire is provided in Appendix A. The questionnaire was piloted by five physicians including three GPs, (two of whom held a university degree in telemedicine). However, they were not members of the expert panel.

Participant’s level of best practices was stratified based on responses to nine questions yielding a total score ranging from 0 to 16 points. The scoring system is detailed in Appendix B. Overall, a good level of knowledge was defined as a score of 12 or higher out of 16.

### 2.2. Statistical Analysis

Answers to the survey were described as mean ± standard deviation or median (range) for quantitative variables, and number (%) for categorical variables. The Mann–Whitney U test was used to compare quantitative variables. We built a multivariate logistic regression model including respondent characteristics and variables with a *p* < 0.20 by univariate analysis to investigate the factors associated with proper use of telemedicine. Backward selection was applied. A *p* < 0.05 was considered statistically significant. All analyses were performed using SPSS 21.0^®^ (SPSS, Chicago, IL, USA).

## 3. Results

The questionnaire was distributed to 4500 GPs, of whom 233 responded (5.2% response rate). Among the respondents, 92 (39.5%) were aged between 30 and 40 years old, 117 (50.2%) were female, 181 (77.7%) were working in a group practice, and 144 (61.8%) were located in rural or semi-urban areas.

A total of 218 GPs (93.6%) performed fewer than 15 teleconsultations per week and 106 (45.5%) reported performing between one and five per week. Before the onset of the COVID-19 pandemic, 10.3% of GPs (*n* = 24) performed teleconsultations and this proportion increased to 74.2% (*n* = 173) at the start of the pandemic in France. Over a quarter of GPs (25.3%, *n* = 59) had a self-declared good level of knowledge of telemedicine practices as defined by a score ≥ 12 out of 16 on the knowledge questions and the rules of good practice are of heterogeneous application.

Our study found that while 73% (*n* = 170) of GPs stated that they had seen the patient physically in the last 12 months or already had a patient come to the office after a teleconsultation (*n* = 169) (two markers of good practice in teleconsultation), only 26% (*n* = 61) systematically obtained the patient’s agreement before carrying out a teleconsultation and 69.1% (*n* = 161) used a video channel in less than 25% of teleconsultations. Table 1 presents the answers (%) to the questionnaire used to score GPs on their level of best practices of teleconsultations.

By multivariate analysis, (i) practicing in an urban area (OR: 2.90 [95% confidence interval (CI): 1.08 to 7.82], *p* = 0.04), (ii) performing a teleconsultation at the patient’s initiative (OR: 5.84 [95% CI: 2.37 to 14.40], *p* < 0.001), and (iii) performing more than five teleconsultations per week (OR: 4.09 [95% CI: 1.89 to 8.82], *p* < 0.001) were factors associated with a significantly higher level of telemedicine best practices.

## 4. Discussion

In this study we were seeking to identify factors that could influence best practice guidelines of telemedicine. Based on our evaluation, only one quarter of responding GPs had a good level of best practices. Our results show that GPs working in urban areas who performed more than five teleconsultations per week were more likely to fit telemedicine best practices. This is in line with the previous literature [12]. However, it may be worth noting that even though telemedicine is often touted as an appealing solution to the lack of medical resources [13,14], there may be a double penalty that rural areas suffer, namely the lack of physicians on the one hand and the lack of high-quality broadband access on the other, which is a pre-requisite, as highlighted in Corteloy-Ward et al. (2020) [15]. Therefore, to avoid accentuating inequalities in access to healthcare, it is of importance to consider the technical limitations of geographical territories where they are to be deployed and to better prioritize and improve technological constraints [16]. Moreover, the use of teleconsultations amongst primary care practitioners may have also evolved post-COVID-19, as seen in Brant et al. (2016) where GPs in urban regions were shown to have a general reluctance toward the use of telemedicine in contrast to face-to-face consultations [17].

Existing guidelines from professional societies and the French health authorities concur that the use of video is one quality criterion in teleconsultations [11,18]. In our study, the rate of routine use of video during teleconsultations (>75% of the time) was 22.3%. Despite the fact that healthcare in France requires the use of both video and audio for teleconsultations, these rules were relaxed during the early stages of the pandemic lockdown, therefore communications by audio only were allowed and accepted for reimbursement [19]. This measure undoubtedly affected the rate of video use in our study, although the rate observed here remains nonetheless higher than the 6.4% observed in another similar study by Scott et al. [12]. Moreover, there is ongoing debate about the quality of the doctor–patient relationship during remote consultations [20]; however there may still be a need to continue promoting the wider use of video during teleconsultations [21].

In our study, the number of teleconsultations performed per week showed a correlation to a GP’s level of best practices about telemedicine. Although the COVID-19 pandemic led to an exponential increase in the use of telemedicine services, previous studies showed that HCPs adapted their use of telemedicine even after the lockdowns and integrated teleconsultations permanently into their usual practice alongside with face-to-face consultations [4]. Apart from on-the-job learning, there may be a clear need to develop structured, adapted training programs for HCPs because currently available resources appear to be insufficient to guarantee a consistent high level of quality of care for the continuing deployment of teleconsultations in primary care [22].

Our study’s main limitation was the low participation rate which could have restricted the interpretation of the results. However, the questionnaire was distributed to a great number of participants in comparison to similar studies that were conducted within other medical specialties [23,24]. Furthermore, although our score evaluating the level of best practices was designed by a panel of experts based on relevant literature, it was not externally validated. Other limitations may have been due to the specific conditions of the French healthcare system and reimbursement model for HCPs using video during teleconsultations. Therefore, our findings may not entirely be generalized to systems with different specifications and requirements. Lastly, we did not include data on the equipment used for the teleconsultations by the GPs and this may have provided useful information on the value related to the transfer of voice and video information between a GP and patient.

## 5. Conclusions

Our study aimed to describe the gap to best practices of teleconsultations by private practice GPs. Our results show that, the proportion of GPs with a good level of best practices was low (25.4%). The working environment (urban or rural) and the number of teleconsultations performed per week were associated with it. Our study reveals a need for structured training for GPs providing teleconsultations in the future. It may also be worth to extend this research to neighboring countries to compare results.

## Figures and Tables

**Table 1 ijerph-19-06220-t001:** Survey scores on telemedicine knowledge among general practitioners (GPs) in a region in France in 2021.

Questions		Score *	*n* = 233 (100%)
Q11. Where do you perform teleconsultation?	In the practice	2	172 (74%)
	From home	0	61 (26%)
	In my car	0	
	Other	0	
Q15. Have you ever requested a patient to come in for a face-to-face consultation after a teleconsultation, because the situation requires a clinical examination?	Yes	2	169 (73%)
	No	0	64 (27%)
Q16. Do you systematically obtain the patient’s consent before proceeding with a teleconsultation?	Yes	1	61 (26%)
	No	0	172 (74%)
Q19. For every patient you have seen by teleconsultation, did you see them at least one in the past 12 months in a face-to-face consultation (excluding exceptions)?	Yes	1	170 (73%)
	No	0	63 (27%)
Q23. Do you use teleconsultation software?	Yes	2	114 (49%)
	No	0	119 (51%)
Q25. Currently, what percentage of your teleconsultations use video?	<25%	0	181 (78%)
	25-50%	1	52 (22%)
	50-75%	2	0 (0%)
	>75%	3	0 (0%)
Q26 (a). Regarding the reimbursement of teleconsultations, do you know the billing code?	Yes	1	182 (78%)
	No	0	51 (22%)
Q26 (b). Regarding the reimbursement of teleconsultations, do you know the cost?	Yes	1	173 (74%)
	No	0	60 (26%)
Q27. Do you know the rules for exemption from payment for teleconsultations	Yes	1	148 (64%)
	No	0	85 (36%)
Q31. For the transmission of medical data, what solutions do you use:	An approved teleconsultation solution (such as Doctolib, Odysweb, etc.,)	2	104 (45%)
	Ordinary post	1	61 (26%)
	The patient comes to get their prescription at the practice	1	
	An encrypted data transfer solution	0	68 (29%)
	Email	0	

* Total score ranging from 0 to 16 points. A good level of knowledge was defined as a score of 12 or higher out of 16.

## Data Availability

The data presented in this study are available upon the request to the corresponding author. The data from this study are not accessible to the public due to French legislation which makes the investigators responsible for data processing.

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
