# Peer review of "Gaps to Best Practices for Teleconsultations Performed by General Practitioners: A Descriptive Cross-Sectional Study"

_ijerph, 2022, doi:10.3390/ijerph19106220_

Round 1

Reviewer 1 Report

Thank you for addressing the key points that were made regarding key aspects of your paper

The paper is much improved and the presentation and results are much easier for an international readership to understand.

Reviewer 2 Report

I think that the authors correctly interpreted the reviewer's comments.The article is scientifically credible. The most recent and most up-to-date factor in the article is the topicality of the scientific topic. I believe that quantification is sufficient for the research group. The article should be carefully edited. I believe the article could be relevant to the WHO. I consider that the work is going in the right direction.

This manuscript is a resubmission of an earlier submission. The following is a list of the peer review reports and author responses from that submission.

Round 1

Reviewer 1 Report

The article is interesting and valuable. The article is really important to collegues working in the field. Therfore, in my opinion, the article should be publicate in IJERPH. The article has a correct IMRaD structure. The scientific argument is conducted in a logical manner. The serious lack of the article is lack of figures, charts etc. A quantified statistical analysis was performed. The correct mathematical apparatus was used, but its level is not high. The statistical survey is scientifically reliable. The references are modern.The reviewer did not notice any errors in reasoning in the article. The article shows a significant practical value for doctors who want to provide medical advice with the use of teleconsultation. The issues raised may also be of interest to people related to the development of a methodology for conducting proper medical consultation. The scientific argument is conducted in a logical manner.The article is scientific in nature. In subsequent works, the authors should show the correct methodology of teleconsultation.

Mandatory change:

1.The article should clearly show the research gap in knowledge by referring to important scientific approaches. In this case, the topic raised will be considered important for science.

Minor:

1. It may be important for the article to show the opinions of supporters and opponents of teleconsultation

2. It may be important for WHO to extend the research to neighboring countries and compare the obtained results

3. There is no description of electronics used for teleconsultation at work, including important values related to the transfer of voice and video information between the doctor and the patient.

Reviewer 2 Report

The topic is important. However, the presentation of the results and discussion are not detailed described.

The paper and analysis are just a couple of pages long and therefore the contribution of this paper is not clear.

The number of responses is very low since only 233 answers from 4500 invitations.

The reference section is also limited. It is important to present an in-depth comparison of the proposed results with the past studies available in the state of the art. 

I recommend the authors to submit this paper to a conference.

Reviewer 3 Report

This is an interesting subject area and worthy of investigation.

The paper uses the term "best practice" but it is unclear what is meant by that or whether it is simply following the recommendations of the French National Authority for Health. also the "Level Of Knowledge" which is a key part of the assessment needs a more in depth explanation as it is unclear what knowledge in particular is being assessed and more importantly  the way this is being assessed through the use of a scoring system for example the weighted scoring related to Questions 11, 15 and 19 are difficult to fully understand. Some questions relate to clinical care delivery and others to national reimbursement rules.....

Why remote consultations conducted from home score 0 whereas from a medical centre score 2 seems counter intuitive if all other good practice processes are in place.